# Association between Lifestyle Behaviors and Health-Related Quality of Life in a Sample of Brazilian Adolescents

**DOI:** 10.3390/ijerph17197133

**Published:** 2020-09-29

**Authors:** Bruno Gonçalves Galdino da Costa, Jean-Philippe Chaput, Marcus Vinicius Veber Lopes, Rafael Martins da Costa, Luís Eduardo Argenta Malheiros, Kelly Samara Silva

**Affiliations:** 1Núcleo de Pesquisa em Atividade Física e Saúde, Universidade Federal de Santa Catarina, Florianópolis 88040-900, Brazil; marcusvvl@hotmail.com (M.V.V.L.); rafael.martins.costa@posgrad.ufsc.br (R.M.d.C.); luis.eduardo.a.m@posgrad.ufsc.br (L.E.A.M.); kelly.samara@ufsc.br (K.S.S.); 2Healthy Active Living and Obesity Research Group, Children’s Hospital of Eastern Ontario Research Institute, Ottawa, ON K1H 8L1, Canada; jpchaput@cheo.on.ca

**Keywords:** behavior, exercise, illicit drugs, screen time, sleep, youth

## Abstract

This study aimed to analyze the association between lifestyle behaviors and health-related quality of life (HRQoL) among Brazilian adolescents. We evaluated 739 adolescents (51.0% girls; mean age, 16.4 ± 1.0 years) from the mesoregion Grande Florianópolis, Brazil. Participants were asked to complete an online questionnaire and sex, age, mother’s education, health-related quality of life, physical activity, screen time indicators, sleep duration, diet, cigarette smoking, alcohol drinking, and drug experimentation were retrieved. Health-related quality of life was assessed using the Kidscreen-10 instrument. Measures of body mass and height were taken by trained researchers. Mixed-effects linear regression models were used. Self-reported health-related quality of life was higher in males (β = 3.68, 95%CI: 2.75; 4.61) compared to females, and no association was observed for age and mother’s education level. Practicing sports (β = 1.19, 95%CI: 0.29; 2.08) was associated with better HRQoL, while processed food score (β = −0.45, 95%CI: −0.78; −0.13), working using screen devices for more than 4 h/day (β = −2.38, 95%CI: −4.52; −0.25), having experimented illicit drugs (β = −2.05, 95%CI: −3.20; −0.90), and sleeping less than 8 h/night (β = −1.35, 95%CI: −2.27; −0.43) were unfavorably associated with HRQoL. Non-sport physical activities, unprocessed food, studying, watching videos, playing videogames, using social media, alcohol drinking, and smoking were not associated with health-related quality of life. These findings suggest that promoting sports and adequate sleep, and preventing excessive workloads and the use of drugs among adolescents may be effective strategies to improve HRQoL.

## 1. Introduction

Health-related quality of life (HRQoL) is a holistic measure of health and encompasses a general sense of well-being related to physical, psychological, environmental, and social aspects of health [1,2]. As HRQoL comprises positive subjective aspects of well-being and covers several aspects of what each person considers important in life, such as having adequate relationships with family and peers, feeling fit and well in the physical aspect, and being satisfied with material needs, it is desirable that all population subgroups have adequate HRQoL [3]; however, adolescence often includes physical, behavioral, and psychosocial changes that can negatively affect HRQoL [2,4].

Health-related lifestyle is a determinant of HRQoL [5,6] constituted by many behaviors (e.g., physical activity, screen time, diet, sleep, use of alcohol, tobacco, and other substances) that can positively or negatively affect peoples’ way of life and health [7], and contribute to the protection or risk for early mortality and chronic diseases [8]. Adequate sleep has been associated with better HRQoL [9]. Additionally, recent reviews showed that physical activity [6,10,11] and screen time [12] have been positively and negatively related to HRQoL, respectively, and sports practice has shown to be even better for some of its dimensions [10]. Furthermore, a Mediterranean diet pattern has been related to increased HRQoL scores [6,13], and alcohol [14,15,16], tobacco [17], and illicit drug [15,16] use seem to be negatively related to HRQoL. Many of the lifestyle behaviors, when adopted in an unhealthy way, have responses that may negatively impact health in ways that compromise HRQoL. For example, lack of sleep affects emotional regulation and cognitive performance [18], use of substances is associated with increased anxiety [19], and high use of social media has been associated with social harassment [20] and depressive symptoms [21].

Identifying modifiable lifestyle factors that are related to HRQoL is necessary to plan effective policies and interventions in adolescence since it is the period where different behaviors are adopted and may track into adulthood. Furthermore, most previous studies have focused on one or two [22] behaviors of the lifestyle, such as physical activity [11], sleep [23], or screen time [12], but have not considered the others, which limits our comprehension of what factors are associated independent of the others (e.g., the relationship between sedentary behavior and HRQoL may not be independent of physical activity). Additionally, the majority of the available evidence is from high-income countries [11,12], which differs from low- and middle-income countries in social and cultural aspects that may impact the lifestyle [24] and HRQoL of adolescents. Brazilian adolescents have been shown to be physically inactive [25] while also experimenting and using substances early [26], as well as having the highest prevalence of unhealthy screen time behavior in a comparison of 12 countries [27]. These behaviors have been shown to affect adolescents’ health, including depressive symptoms [28], and their relation with HRQoL should be evaluated. Thus, the present study aimed to analyze the association between lifestyle behaviors and HRQoL in a sample of Brazilian adolescents while considering other lifestyle behaviors in the analyses.

## 2. Methods

### 2.1. Study and Participants

This study analyzed data from the baseline sample of the Estudo Longitudinal do Estilo de Vida de Adolescentes (ELEVA). All public schools (*n* = 3) offering high school courses integrated with professional courses in the mesoregion Grande Florianópolis, Southern Brazil, were invited and joined the study. A census method was adopted, and all students present in the classrooms during data collection (August to December 2019) were invited to participate. A minimum of three visits were conducted for each class. Individuals who had physical and/or mental limitations that prevented participation in the study measurements were not included. Of all the students enrolled in the lists provided by the schools (*n* = 1618), 1249 were present on at least one day of data collection and were invited to participate. Participants received ascent and consent forms for them and their legal guardians to sign, respectively. From those, 1010 participants provided signed forms and were included in the study for further procedures. The present research project was approved by the Ethics Committee in Research with Human Beings of the Federal University of Santa Catarina (protocol number: 3.168.745).

### 2.2. Procedures

Participants were asked to complete an online questionnaire hosted on the SurveyMonkey platform (SVMK Inc, San Mateo, CA, USA). Trained researchers explained the questionnaire and provided smartphones, tablets, and/or laptop computers for the students to use, or they could use their device if they chose to. The questionnaire was applied during class time, and researchers helped with questions and/or technical difficulties (e.g., devices with low battery, unstable internet connection). The average completion time was 24 min. Measures of body mass (kg) and height (cm) were taken by trained researchers, in a private room, with students wearing light clothes during class time. Body mass was measured using a balance calibrated to the nearest 0.1 kg (Welmy^®^). A portable stadiometer (Alturaexata^®^) was used to measure stature to the nearest 0.1 cm.

### 2.3. Measures

#### 2.3.1. Outcome Variable

HRQoL was assessed using the Kidscreen-10 [1,2]. This instrument consists of a 10-item questionnaire, which reflects the respondents’ perceptions and feelings. This instrument has been validated for the Brazilian adolescents [29]. Its psychometric properties in adolescents aged 12 to 18 years are considered adequate, with Cronbach’s alpha of 0.81, test and re-test intraclass correlation of 0.69, and correlations between Kidscreen-10 score with the 5 dimensions of the Kidsreen-52 ranging between 0.26 and 0.72 [2]. The answers are coded into a score that ranges from 0 to 100, with higher scores representing better HRQoL.

#### 2.3.2. Correlates

Physical activity was measured with the Self-Administered Physical Activity Checklist [30], which is a list with 22 activities, where the participants register the frequency (0–7 days/week) and duration (min/day) they engage in each of these activities in a habitual week. Students could include other activities not included in the pre-established list of activities or answer that they do not engage in any physical activity at all. This questionnaire has been validated and used with Brazilian adolescents [31]. Two variables were calculated, i.e., the volume of sports and non-sports activities (in minutes/week) [10]. The weekly volume of sports was calculated by summing the volume of soccer, futsal, basketball, handball, volleyball, tennis, table tennis, swimming, athletics, martial arts, rhythmic gymnastics, cycling, skating/skateboarding, and surfing, and the volume of non-sports included capoeira, dancing, gym gymnastics, resistance training, walking, jogging/running, and active playing.

Screen time was measured across five activities: studying; working; watching videos; playing games; and using social media/chat apps. Students reported how many hours and minutes a day they spent on each of those activities when using any screen-based device during weekdays and weekend days. The volume on each activity was weighted ([volume in weekdays × 5 + volume in weekend days × 2]/7), and classified into less than two hours/day, between two and four hours/day, and more than four hours a day. These items were validated in a previous pilot study (*n* = 104 adolescents) [32], yielding Gwet’s AC_2_ agreement coefficients ranging from 0.54 to 0.82 for weekdays, and 0.56 to 0.87 for weekend days.

Sleep duration was calculated by the difference between self-reported bedtimes and wake-up times (hours and minutes) separately for weekdays and weekend days. The volume of sleep duration was estimated and weighted using the following formula: [(volume in weekdays × 5 + volume in weekend days × 2]/7). After this procedure, participants were categorized according to sleep duration recommendations: healthy sleepers (8–10 h/day), short sleepers (<8 h/day), and long sleepers (>10 h/day) [33].

Diet was measured using the following question: ‘In the last seven days, how many days did you consume…?’ regarding each one of the following foods: beans, vegetables, fresh fruits, fried salty foods (e.g., French fries, fried chicken), sweets (e.g., candies, bubble gum), processed foods (e.g., ham, chicken nuggets, instant noodles), fast food, and soda. The answers ranged from zero to seven days, and the portion size was not reported. These questions have been validated for Brazilian adolescents [34]. The items were reduced using principal component analysis, using the criteria of eigenvalue >1.0, with orthogonal Varimax. The model had Kaiser–Meyer–Olkin (KMO) criteria of 0.70, which was deemed adequate. The final model was composed of two components, with the first one heavily loaded on sweets, fried salty foods, processed goods, and fast food, and was named ‘processed foods’, and the second was characterized by high loadings on fresh fruits and vegetables, and was named ‘unprocessed foods’.

The frequency of cigarette smoking and alcohol drinking over the past 30 days was obtained, and participants who smoked at least once, and drank alcohol at least once were classified as smokers (vs. non-smokers) and drinkers (vs. non-drinkers), respectively. Drug experimentation was assessed using a question asking if the participant ever tried an illicit drug (e.g., cannabis, cocaine, crack, LSD), and those who reported ‘yes’ were classified as having experimented.

#### 2.3.3. Adjusting Variables

The body mass index was calculated, and participants were classified using the World Health Organization growth curves (severe thinness, thinness, normal weight, overweight, obesity) [35]. Participants in the severe thinness and thinness were pooled together (*n* = 3). Participants also reported their sex (boys/girls), age (in completed years), and mother’s education (<8 years, 8–11 years, >11 years, unknown).

### 2.4. Statistical Analyses

The characteristics of the sample are presented using means and standard deviations, and absolute and relative frequencies for continuous and categorical data, respectively. To test the associations between lifestyle behaviors and HRQoL, the Kidscreen-10 score was used as the outcome variable in mixed-effects linear regression models. For the crude analyses, the association of each lifestyle behavior with HRQoL was tested in a model adjusted for sex, age, mother’s education, and weight status. Sex interaction terms (e.g., sex*use of alcohol, sex*videogames) were tested for each indicator, as previous work has found that the relationship between some behaviors and HRQoL may differ according to sex [10]; however, as no significant interactions were observed, the reported models are with the total sample. A model with all lifestyle behaviors and adjusting variables was fit, but due to the increased number of parameters, it was used as a reference for fitting. A series of models were fit according to a backward procedure and were compared using goodness of fit parameters (i.e., Akaike information criterion, Bayesian information criterion). The reduced model did not differ from the model with all variables when compared by the Wald test. Thus, the most parsimonious model, which included all variables associated with HRQoL after mutual adjustment, was chosen. All analyses considered the hierarchical structure of participants nested within schools. Analyses were conducted in R, version 4.0 for Windows (R Foundation for Statistical Computing, Vienna, Austria), using the package “lme4”, version 1.1-23.

## 3. Results

A total of 1010 participants provided written informed consent, 856 answered the questionnaire, and 851 had valid measures of body mass and height, of which 739 (73%) provided valid information on all included variables (e.g., the questionnaire was not incomplete). The characteristics of the sample can be observed in Table 1. Briefly, 51% of the sample were girls, and the mean age was 16.4 ± 1.0 years, 52.4% had mothers that studied for more than 11 years, and participants had an average HRQoL score of 40.6 ± 6.6 using the Kidscreen-10 tool.

The association between lifestyle behaviors with HRQoL is displayed in Table 2. In the adjusted model, the association of sports (β = 1.19, 95%CI 0.29; 2.08), processed food score (β = −0.45, 95%CI −0.78; −0.13), working using screen devices for more than 4 h/day (β= −2.38, 95%CI −4.52; −0.25) when compared to less than 2 h/day, having experimented illicit drugs (β= −2.05, 95%CI −3.20; −0.90), and sleeping less than 8 h/night rather than 8–10 h/night (β= −1.35, 95%CI −2.27; −0.43) were significantly associated with HRQoL. Associations between the use of screens for recreational purposes (i.e., watching videos, playing videogames, and using social media) and having smoked at least one tobacco cigarette in the last 30 days with HRQoL were only observed in crude models but not in the adjusted models, and thus were not included in the presented adjusted model.

## 4. Discussion

This study reports the association between lifestyle behaviors and HRQoL in a sample of Brazilian high school adolescents. In the initial analyses, we observed that a lower volume of sports, eating processed foods, working on a screen device, watching videos, playing videogames, using social media, having experimented with illicit drugs, smoking, and sleeping insufficiently were associated with lower HRQoL. However, after adjustments for other behaviors, we observed that only the volume of sports, eating processed foods, working on a screen device, having experimented with illicit drugs, and sleeping insufficiently were significantly associated with lower HRQoL independently of other behaviors. This suggests that multiple lifestyle factors are important when HRQoL is concerned, and from a holistic approach, interventions targeting multiple behaviors may be needed to improve the well-being of adolescents. We believe the findings of this study expand our understanding of the etiology of HRQoL in adolescents of middle-income countries and provide insights for possible interventions.

The relationship between physical activity and HRQoL in adolescents has been shown in a previous review [11]; however, the effect sizes for cross-sectional studies are small to negligible. One hypothesis is that some activities, such as sports, are more important for HRQoL than others (e.g., doing chores), which is supported by a recent study in adolescents that showed that sports activities were more strongly related to HRQoL than non-sport activities [10]. These results are similar to the ones observed in the present study that despite a small effect size, may be important if participants engage in it an hour every day. It is important to notice that this relationship was still significant after adjustment for other lifestyle behaviors, suggesting that increasing participation in sports may be an effective way to increase HRQoL even when other behaviors are not changed. This could improve interventions based on physical activity for the improvement of HRQoL in pediatric populations, which so far have shown modest effects [11].

Findings regarding screen time behaviors suggest that only working for more than four hours a day remained associated with lower HRQoL after adjustment for other lifestyle behaviors. This finding suggests that it may not be the screen time that negatively affects HRQoL but rather the increased workloads, as participants who have to work for longer hours, combined with school homework, may have less leisure time available compared to those who do not work.

In relation to sleep, a previous study with 9–11-year-olds from 12 countries also found no association between accelerometer-measured sleep indicators with HRQoL assessed with the Kidscreen 10 [36]. Concerning sleep, short sleepers had lower scores of HRQoL compared to healthy sleepers in the present study. This finding is similar to other studies with adolescent samples [9,23,36]. Sleep is an important behavior for many body systems, and poor sleep has been associated with impaired emotional regulation and cognitive performance [18], which can be a pathway that at least partly explains the relationship with HRQoL.

Eating processed foods was related to lower HRQoL in the present study, whereas the unprocessed food score was not. The relationship between diet and HRQoL has been studied before, and a Mediterranean diet has been associated with better scores of HRQoL in adolescents [6,13]. In the present study, we adopted a data-driven approach to identify dietary patterns, which revealed two scores that reflected processed and unprocessed foods, and do not directly compare to adherence to the Mediterranean diet. However, it is important to notice that the low consumption of processed foods is a defining characteristic of the Mediterranean diet, is associated with better health outcomes [37], and is recommended by current dietary guidelines for the Brazilian population [38]. Increased consumption of ultra-processed foods, in particular, has been shown to be associated with many unhealthy outcomes, such as obesity, diabetes, and cancer [33,37], and although its mechanisms and pathways are not clear, this may partially explain its relationship with HRQoL observed in the present study.

Drug experimentation was associated with lower HRQoL in the present sample, as observed in adolescent samples in previous studies [15,16]. This relationship is not clear, and one possible explanation is that other non-observed variables predispose adolescents to health-risking behaviors (e.g., experimenting with drugs, aggressive behavior, high-risk sexual behavior) that are also associated with lower HRQoL, but some were not investigated in the present study [15]. This hypothesis, however, would suggest that alcohol and tobacco are associated with HRQoL, which was not observed in the present study. Additionally, adolescents with low HRQoL may try substances in a way to cope with the problems, but, to confirm this, prospective studies are needed.

Although some lifestyle behaviors, such as the use of social media and cigarette smoking, were observed to be associated with HRQoL in the crude models, the association was no longer statistically significant after mutual adjustment with other behaviors. Previous studies have shown that some of these behaviors were associated with HRQoL in adolescent samples; however, this relationship is not necessarily independent of other behaviors (e.g., the association between social media and HRQoL may not be independent of physical activity), and in many of these studies, independence may have not been tested by including several lifestyle behaviors in the analytical models. Future longitudinal studies could adopt structural equation modeling techniques to check if changes in lifestyle behaviors are related to each other and impact HRQoL. Overall, the results of the present study suggest that policies and interventions should target multiple behaviors, including sleep, diet, physical activity, and the prevention of drug use, to increase adolescents’ HRQoL.

Whereas no significant sex interaction terms were observed for the associations of lifestyle behaviors and HRQoL, boys had higher scores compared to girls. The sex difference observed in the present study is consistent with current literature [10,15,39], with girls being more susceptible to perceive their health as poor, having functional limitations, increased depressive symptoms, and low self-esteem [39]. Girls may have increased social pressures and hormonal changes that increase their sensibility to stress during adolescence [40], which predisposes them to depression as well. Additionally, whereas no interactions were observed, physical activity was related to HRQoL in this and previous studies [10,22], and girls systematically engage in less physical activity compared to boys [41]. This illustrates that intervening in lifestyle behaviors among girls may also provide additional benefits that can improve HRQoL.

The findings of the present study suggest that several lifestyle factors are related to adolescents’ HRQoL. Our findings can be useful in the planning of policies, and interventions should be taken into account in future studies. For example, intervention studies or school managers aiming at improving HRQoL of adolescents may have success if improvements in sports participation (e.g., offering classes) and reduction of the ingestion of processed food (e.g., changing cafeteria policies) occur. Additionally, longitudinal and experimental studies are needed to confirm the direction of the associations observed in our study. Another important aspect to be considered in future research is to explore how other variables, such as self-esteem, environmental factors, and family and peer relations, can influence HRQoL and may interact with lifestyle behaviors.

This study has limitations to be acknowledged. The sample was small in relation to the large scope of associated exposures. Self-report instruments were used to measure habitual behaviors, which may be prone to recall limitations and social desirability bias. However, qualitative aspects of behaviors were explored, including the types of screen time behaviors and physical activity, which are still a challenge to be objectively measured. A single overall measure of HRQoL was evaluated in this study, which limited the understanding of findings, as HRQoL is a complex latent construct that can be analyzed in distinct dimensions by instruments with a higher number of items. The main strength of this study was the provision of a comprehensive analysis of the associations between several lifestyle behaviors and HRQoL by exploring qualitative aspects of measured behaviors, and among adolescents of a middle-income country, while taking into account the other behaviors in our analyses.

## 5. Conclusions

This study identified that adolescents who spent more time on sports but not on non-sports physical activities had higher scores of HRQoL. The time spent in work-related screen activities was inversely associated with HRQoL score. However, this association was not observed for recreational screen time indicators (i.e., watching videos, playing videogames, or using social media). In addition, girls, adolescents who experimented with drugs in their life course, those with higher consumption of ultra-processed foods, and those who sleep insufficiently reported lower scores of HRQoL. These findings suggest that promoting sports and adequate sleep, and preventing the use of drugs and excessive workloads among adolescents may be effective strategies to improve HRQoL.

## Figures and Tables

**Table 1 ijerph-17-07133-t001:** Characteristics of the sample of Brazilian adolescents, 2019.

Variable	Total (*n* = 739)	
	Mean	SD
**Health-related Quality of life (0–100 score)**	40.6	6.6
**Age (years)**	16.4	1.0
**Physical activity**		
Volume of sports (hours/day)	0.3	0.5
Volume of non-sports (hours/day)	0.3	0.5
**Diet**		
Unprocessed (score)	3.22	1.24
Processed (score)	2.79	1.38
	*n*	%
**Sex**		
Girls	378	51.1%
Boys	361	48.9%
**Mother’s education**		
<8 years	78	10.2
8–11 years	263	34.4
>11 years	400	52.4
Unknown	23	3
**Screen time behaviors**		
* Studying*		
<2 h/day	455	59.6
2–4 h/day	193	25.3
>4 h/day	116	15.2
* Working*		
<2 h/day	646	84.6
2–4 h/day	83	10.9
>4 h/day	35	4.6
* Watching videos*		
<2 h/day	274	35.9
2–4 h/day	299	39.1
>4 h/day	191	25
* Playing games*		
<2 h/day	517	67.7
2–4 h/day	127	16.6
>4 h/day	120	15.7
* Using social media*		
<2 h/day	267	34.9
2–4 h/day	219	28.7
>4 h/day	278	36.4
**Drugs**		
Non-experimenter	619	81
Experimenter	145	19
**Alcohol**		
Non-drinker	435	56.9
Drinker	329	43.1
**Tobacco smoking**		
Non-smoker	705	92.3
Smoker	59	7.7
**Sleep duration**		
Healthy sleepers (8–10 h/night)	331	44.8
Short sleepers (<8 h/night)	363	49.1
Long sleepers (>10 h/night)	45	6.1
**Weight Status**		
Normal weight	574	75.2
Overweight	134	17.5
Obesity	56	7.3

**Table 2 ijerph-17-07133-t002:** Association between lifestyle indicators and health-related quality of life in a sample of Brazilian adolescents (*n* = 739), 2019.

Variable	Health-Related Quality of Life Score (0–100)	
	Crude	Adjusted
	β (95%CI)	β (95%CI)
**Sex**		
Girls	Reference	Reference
Boys	**4.24 (3.33; 5.15)**	**3.68 (2.75; 4.61)**
**Age (years)**	−0.18 (−0.61; 0.25)	-
**Mother’s education**		
<8 years	Reference	Reference
8–11 years	0.25 (−1.36; 1.85)	0.03 (−1.55; 1.60)
>11 years	**1.70 (0.15; 3.25)**	1.37 (−0.15; 2.89)
Unknown	−0.47 (−3.50; 2.55)	−0.78 (−3.73; 2.16)
**Physical activity**		
Volume of sports (hours/day)	**1.23 (0.30; 2.16)**	**1.19 (0.29; 2.08)**
Volume of non-sports (hours/day)	0.16 (−0.74; 1.06)	-
**Diet**		
Unprocessed (score)	0.23 (−0.14; 0.61)	-
Processed (score)	**−0.55 (−0.88; −0.22)**	**−0.45 (−0.78; −0.13)**
**Screen time behaviors**		
*Studying*		
<2 h/day	Reference	Reference
2–4 h/day	0.43 (−0.64; 1.50)	-
>4 h/day	0.01 (−1.30; 1.33)	-
*Working*		
<2 h/day	Reference	Reference
2–4 h/day	−0.99 (−2.42; 0.43)	−0.68 (−2.07; 0.72)
>4 h/day	**−3.09 (−5.27; −0.91)**	**−2.38 (−4.52; −0.25)**
*Watching videos*		
<2 h/day	Reference	Reference
2–4 h/day	−0.13 (−1.16; 0.91)	-
>4 h/day	**−1.28 (−2.45; −0.11)**	-
*Playing games*		
<2 h/day	Reference	Reference
2–4 h/day	−0.53 (−1.83; 0.76)	-
>4 h/day	**−1.50 (−2.86; −0.14)**	-
*Using social media*		
<2 h/day	Reference	Reference
2–4 h/day	−0.44 (−1.57; 0.69)	-
>4 h/day	**−1.69 (−2.78; −0.60)**	-
**Drugs**		
Non-experimenter	Reference	Reference
Experimenter	**−2.28 (−3.45; −1.11)**	**−2.05 (−3.20; −0.90)**
**Alcohol**		
Non-drinker	Reference	Reference
Drinker	−0.71 (−1.62; 0.21)	-
**Tobacco smoking**		
Non-smoker	Reference	Reference
Smoker	**−2.30 (−4.03; −0.57)**	-
**Sleep duration**		
Healthy sleepers (8–10 h/night)	Reference	Reference
Short sleepers (<8 h/night)	**−1.66 (−2.59; −0.73)**	**−1.35 (−2.27; −0.43)**
Long sleepers (>10 h/night)	−1.77 (−3.70; 0.15)	−1.43 (−3.32; 0.45)

Note. β (95%CI): Regression coefficients and 95% confidence intervals (in HRQoL score). The crude models included sex, age, body weight status, and mother’s education. The adjusted model was selected by comparing adjusted models using fit parameters (Akaike information criterion, Bayesian information criterion). -: Variables were not included in the adjusted model. Values in bold indicate statistical significance at *p* < 0.05.

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
