# Peer review of "Association between Lifestyle Behaviors and Health-Related Quality of Life in a Sample of Brazilian Adolescents"

_ijerph, 2020, doi:10.3390/ijerph17197133_

Round 1
Reviewer 1 Report
The authors analyze the association between lifestyle behaviors and health-related quality of life in a sample of three public high school schools in Brazil. It is a subject already studied by other researchers in other countries. However, in this study, new variables not previously taken are added, such as the consumption of harmful substances and the sleep pattern. Also, it was conducted in a middle-income country, unlike other studios. They found that practicing sports, processed food, working using screen devices for more than 4 h/day, having experimented with illicit drugs, and sleeping less than 8 h/night were significantly associated with health-related quality of life. The study has its limitations, which are mentioned by the authors.
My comments are:
1) In section 2.3.1, the references are not in the proper format. Furthermore, reference 21 may not correspond to the text.
2) In section 2.3.2, the Self-Administered Physical Activity Checklist instrument does not have the corresponding reference. Add the references to all the instruments, data, and studies consulted and used.
3) The tables are broken down into two pages.
4) Please add details of the pilot study conducted.
5) It is suggested to include differences between girls and boys in the abstract and the conclusion.
6) It is suggested to include other variables related to self-esteem and family and social environment in the study in future work.
Reviewer 2 Report
Overall, this is a well described, analyzed, and reported research study. The methodology is sound and the research instruments used have good psychometric properties.
Questions/Feedback/comments:
Why was the study done? A bit more background on the general health status of Brazilian adolescents that prompted the study would provide better context for the purpose of the study as well as the questions posed in the research, and how the findings can be used in practice, programming, and policy making specific to addressing the wellbeing of Brazilian adolescents.
Reviewer 3 Report
The Authors present a paper: " Association between lifestyle behaviors and health-related quality of life in a sample of Brazilian adolescents" where they through an online questionnaire identified some correlated findings suggesting that promoting sports and adequate sleep, preventing the use of drugs and excessive workloads among adolescents may be effective strategies to improve health-related quality of life. The manuscript completes previous studies including important and interesting correlation. The message from Authors is interesting from am educational point of view, the sample they studied as well the adopted methodology are fine to me. Also the statistical analysis is appropriate. Nevertheless I suggest to adjust something to make the paper more applicable and understandable:
- When they consider the body mass of adolescents taken by trained researchers, the family body constitution has been considered ? Please specify
- table 1 and 2 could be condensed reporting only the statistical significance. In the text could be reported the detailed analysis.
- in the discussion a summary Flow chart indicating lower or high HRQL for each item should be included (sports, working, sleep, eating, drug alcohol and tobacco...) for having a more immediate view of benefits or not
- In the Conclusion I would appreciate have a feedback from Authors (as suggestion) on how to make workable and applicable their suggestions/conclusions
Reviewer 4 Report
The authors state that the study aims to analyze the association between lifestyle behaviors and health-related quality of life among Brazilian adolescents. To achieve the proposed objective, 739 adolescents from the mesoregion Grande Florianópolis, Brazil was evaluated using a questionnaire. Participants were asked to complete an online questionnaire and sex, age, mother’s education, health-related quality of life, physical activity, screen time indicators, sleep duration, diet, cigarette smoking, alcohol drinking, and drug experimentation were retrieved. The results suggest that Non-sport physical activities, unprocessed food, studying, watching videos, playing videogames, using social media, drinking, and smoking were not associated with health-related quality of life. These findings suggest that promoting sports and adequate sleep, preventing drugs, and excessive workloads among adolescents may be effective strategies to improve health-related quality of life.
The topic of your paper is very interesting and might be of interest to readers. You have also presented an interesting framework. I have a few suggestions:
1 - What was the period that the study was carried out?
2 - What was the criterion for selection and sample size?
3- What are the practical, theoretical, and political implications of the study?
4 - Add suggestions for future research.
Round 2
Reviewer 3 Report
I consider the answers to my request exhaustive enough and explicative.